# Effect of Neuroendocrine Neoplasm Treatment on Human Reproductive Health and Sexual Function

**DOI:** 10.3390/jcm11143983

**Published:** 2022-07-08

**Authors:** Virginia Zamponi, Anna La Salvia, Maria Grazia Tarsitano, Nevena Mikovic, Maria Rinzivillo, Francesco Panzuto, Elisa Giannetta, Antongiulio Faggiano, Rossella Mazzilli

**Affiliations:** 1Department of Clinical and Molecular Medicine, ENETS Center of Excellence, Sapienza University of Rome, 00185 Rome, Italy; virginia22.zamponi@gmail.com (V.Z.); nev.mikovic@gmail.com (N.M.); antongiulio.faggiano@uniroma1.it (A.F.); rossella.mazzilli@uniroma1.it (R.M.); 2Medical Oncology 2, IRCCS Regina Elena National Cancer Institute, 00144 Rome, Italy; 3Department of Medical and Surgical Science, University Magna Graecia, 88100 Catanzaro, Italy; mariagrazia.tarsitano@unicz.it; 4Digestive Disease Unit, ENETS Center of Excellence, Sant’Andrea University Hospital, 00189 Rome, Italy; mrinzivillo@ospedalesantandrea.it (M.R.); francesco.panzuto@uniroma1.it (F.P.); 5Department of Medical-Surgical Sciences and Translational Medicine, Sapienza University of Rome, 00185 Rome, Italy; 6Department of Experimental Medicine, Sapienza University of Rome, 00185 Rome, Italy; elisa.giannetta@uniroma1.it

**Keywords:** fertility, sexual dysfunction, sexual function, neuroendocrine tumors, neuroendocrine neoplasms, treatment, QoL

## Abstract

Neuroendocrine neoplasms (NEN) are characterized by a wide clinical heterogeneity and biological variability, with slow progression and long survival in most cases. Although these tumors can affect young adults, there are few studies that focus on the sexual and reproductive system. The aim of this review was to evaluate the effect of NEN treatment, including somatostatin analogues (SSA), targeted therapy (Everolimus and Sunitinib), radiolabeled-SSA and chemotherapy, on male and female reproductive systems and sexual function. This narrative review was performed for all available prospective and retrospective studies, case reports and review articles published up to March 2022 in PubMed. To date, few data are available on the impact of SSA on human fertility and most of studies come from acromegalic patients. However, SSAs seem to cross the blood–placental barrier; therefore, pregnancy planning is strongly recommended. Furthermore, the effect of targeted therapy on reproductive function is still undefined. Conversely, chemotherapy has a well-known negative impact on male and female fertility. The effect of temozolomide on reproductive function is still undefined, even if changes in semen parameters after the treatment have been described. Finally, very few data are available on the sexual function of NEN treatment.

## 1. Introduction

Neuroendocrine neoplasms (NENs) are a relatively rare and complex group of neoplasms that originate from the cells of the diffused neuroendocrine system, with an incidence of 6.98/100,000 [1]. NENs are characterized by a wide clinical heterogeneity of site of origin (i.e., gastroenteropatic, pulmonary, etc.), biological variability and generally long survival and slow progression. Somatostatin analogues (SSA), octreotide and lanreotide represent the main systemic treatments for advanced well-differentiated neuroendocrine tumors (NET). Regarding the other treatment options for the management of unresectable or metastatic NETs, targeted therapy (Everolimus and Sunitinib), radiolabeled-SSA such as the β-emitter 177Lu-DOTA-D-Phe-Tyr3-octreotate (177Lu-oxodotreotide or 177Lu-DOTATATE) for peptide receptor radionuclide therapy (PRRT) and chemotherapy have been approved [2]. These also represent the standard of care for patients with neuroendocrine carcinomas (NEC) [3].

Recent evidence suggests that both clinical severity and heterogeneity of NENs have a negative impact on the patient’s quality of life (QoL) [4]. However, although sexual health should be considered an essential component of QoL, few studies have explored sexual function and dysfunction in patients with NENs, with non-univocal results.

Furthermore, many NETs have been described in the context of heritable tumor syndromes, such as multiple endocrine neoplasia types 1 and 2 and Von Hippel-Lindau disease, involving young subjects [5]. Over the past decades, many studies have highlighted the importance of reproductive health and the preservation of fertility in cancer patients, mainly in young patients, with the development of the oncofertility, defined as the study of interactions between cancer, anticancer treatment and reproductive health [6]. However, little is known regarding the effects of the treatment in patients with NEN.

The aim of this review was to evaluate the effect of NEN treatment, including SSA, PRRT, targeted therapy and chemotherapy, on male and female reproductive systems and sexual function.

## 2. Materials and Methods

This narrative review was performed for all available prospective and retrospective studies, case reports and review articles published up to March 2022 in PubMed. Data were extracted from the text and from the tables of the manuscript. The keyword search used included “female fertility”, “male fertility”, and “sexual dysfunction” plus: “neuroendocrine tumors”, “neuroendocrine neoplasm”, “carcinoid”, “neuroendocrine tumors and Somatostatin analogues”, “neuroendocrine tumors and SSA”, “neuroendocrine tumors and octreotide”, “neuroendocrine tumors and lanreotide”, “neuroendocrine tumors and targeted therapy”, “neuroendocrine tumors and everolimus”, “neuroendocrine tumors and sunitinib”, “neuroendocrine tumors and radiolabeled-SSA”, “neuroendocrine tumors and peptide receptor radionuclide therapy”, “neuroendocrine tumors and PRRT”, “neuroendocrine tumors and chemotherapy”, and “neuroendocrine carcinomas and chemotherapy”.

## 3. Results

### 3.1. NET Treatment and Fertility

#### 3.1.1. Somatostatin Analogues

Somatostatin (SST) is a polypeptide hormone existing in two forms (14 and 28 amino acids). The properties of SST consist of antisecretory, antiproliferative and antiangiogenic effects [7]. SST has a short half-life and binds with high affinity to five different subtypes of receptor ubiquitously distributed. The binding of SST with its receptor determines several biological functions [7]. The expression levels of SST receptors (SSTRs) have been determined in multiple human tissues across the human body, including the brain, gastrointestinal tract, pancreas, lung and genitourinary tract.

SST is involved in several physiological functions. In the central nervous system, it is synthesized in the serotoninergic neurons of the raphe nuclei, acting in the regulation of mood, sexuality, anxiety, sleep, appetite and body temperature. At the peripheral level, SST is synthesized by specialized enteroendocrine cells located in the gastrointestinal tract, with a role in modulating gut motility. SSTRs are present both in normal and in tumor tissues which enables their response to applied SST analogs (SSA). SSAs have a longer half-life than SST and they have a similar STTR binding profile, with high SSTR2 and moderate SSTR5 affinity. SSAs are very effective drugs for hormonal syndrome control in functioning tumors and exert an antiproliferative effect by inducing cell cycle arrest and apoptosis, and through immunomodulatory effects and angiogenesis inhibition. Considering these data, they have been approved for the treatment of acromegaly and NET [7,8]. SSAs allow the control of cell proliferation through two different mechanisms: a direct one, consisting of the binding to specific surface receptors, and an indirect one, consisting of the inhibition of growth factors and modulation of immune response [7]. If the suppressive effect of SSA on GH and IGF1 is well established, those on the prolactin, luteinizing hormone (LH) and follicular-stimulating hormone (FSH) are still unclear [8].

Few data are available on the possible impact of SSA on fertility. Considering the male reproductive system, Sasaki et al. demonstrated for the first time the presence of SST in human testicular extracts [9]. Subsequent studies on animal models demonstrated the presence of all SSTRs in the testis and their possible impact on the development of Sertoli cells and on spermatogenesis [10,11]. Specifically, Riaz et al. highlighted an overexpression of SSTR2 in a rat’s testis [11]. In the same study, the authors demonstrated that SSTR2 and SSTR5 receptors are crucial during the Sertoli cell developmental period [11]. These data are also confirmed in porcine models [12]. In this regard, Goddard et al. demonstrated the presence of SST2A in Sertoli cells and in spermatogonia in immature porcine testes [12]. These studies also showed that the treatment with SST promotes the apoptosis of Sertoli cells and impaired spermatogonial development, interrupting them in G1 phase of the cell cycle through the inhibition of FSH and stem cell factor expression in Sertoli cells [11,12]. Furthermore, the administration of SST induces a dose-dependent RNA suppression of the kitl gene, which is involved in the regulation of spermatogenesis [11]. In human testes, SSTR1 SSTR2 and SSTR5 were identified [13]. In particular, SSTR1 was detected in Leydig cells, SSTR2 was found in the basal part of the tubules (Sertoli cells) while SSTR5 in the luminal part, suggesting a possible role of SSA and SSTR in human germ cell development, similar to what was described in animal models [13] (Table 1). Moreover, GH plays a key role in gonadotropin secretion and tissue reactivity in men. This way, it supports gonadal differentiation, steroidogenesis and gametogenesis [14]. SST has an inhibitory action on GH secretion; therefore, it has been hypothesized that in studies conducted on males with pituitary adenomas, SSA played a role in the impairment of male fertility [15] (Figure 1).

Considering female fertility, previous observations highlighted the importance of the GH axis for the latest stage of puberty and start of fertility [16,17]. GH could promote follicular maturation and make granulosa cells responsive to gonadotropins via the IGF-1 pathway [15]. The first ovulation seems to be facilitated by GH while it is delayed by SSAs, since they reduce GH secretion. These data demonstrate that any factor (such as SSA) that affects GH secretion during puberty could have deleterious effects not only on growth but also on the onset of fertility [18] (Figure 2).

Few data are available on the safety of SSA in pregnancy and studies are mainly focused on acromegaly. SSA would cross by passive diffusion the blood–placental barrier and bind all five subtypes of SSTRs of the placenta. SSAs have been found in various maternal–fetal fluids, although the precise amount crossing the placenta has not yet been clearly quantified in humans, especially concerning octreotide [19] (Figure 2).

Although most cases of SSA exposure during pregnancy have not been associated with negative maternal–fetal outcomes, some cases of fetal growth retardation and low birth weight have been reported [19]. Octreotide promotes a vasodilation action on the splanchnic circulation [20]. Studies have shown a transient reduction in uterine artery flow immediately following octreotide administration [19]. Fewer data are available concerning the pregnancy safety of lanreotide than octreotide. For this reason, the FDA has included them in different categories in the classification of risk in pregnancy: Lanreotide is in class C, which means that adverse effects on the fetus were detected in animal studies, but inadequate and uncontrolled studies were conducted for humans. Additionally, octreotide is in class B, which means that animal studies failed to demonstrate a risk to the fetus and inadequate studies on pregnant women are available [19,21].

Considering these premises, the guidelines of the Endocrine Society suggest replacing long-acting SSA with short-acting SSA, starting two months before conception [22]. In the case of ascertained pregnancy in a patient with acromegaly, the suspension of SSA is recommended, except in cases where there are symptoms or a lack of control over the tumor [22].

In women with NET, pregnancy can impair the hormonal environment and influence the progression of the tumors or symptoms. In the patients with Carcinoid Syndrome, SSA therapy must be carefully considered to avoid cardiac complications. On the other hand, therapy with SSA could suppress placental production of serotonin and impair fetal neurogenesis [23,24]. For all these reasons, pregnancy planning is strongly recommended in women with NET [23].

#### 3.1.2. Peptide Receptor Radionuclide Therapy

PRRT is indicated in patients with metastasized or inoperable somatostatin receptor-positive NETs. The currently most used radioligands are [90Y-DOTA0,Tyr3] octreotide and [177Lu-DOTA0,Tyr3] octreotate. PRRT is mostly associated with short-term adverse effects [25]. Although some endocrine organs express SST receptors, it is still unclear whether PRRT can lead to hormonal imbalances [26]. In normal tissues, the density of SST receptors is lower than in NETs but it does not exclude that radiosensitive organs, such as gonads, can be damaged by radiomarked-SSA systemically administered. In the study of Teunissen et al., FSH and Inhibin B underwent significant variations after three months from PRRT, suggesting a transient impairment of spermatogenesis. The hormonal imbalance tended to restore in 24 months. A reduction in total testosterone and SHBG at 3 months and a further decline at 24 months with a concomitant increase in the LH level was also found [25]. Few data are available about the impact of PRRT on female fertility. Zhang et al. reported a case report of a young women with an ovarian NET who underwent four cycles of PRRT. In total, 67 months after four cycles of PRRT, she naturally conceived and delivered a healthy baby [27]. Despite the authors of this case report suggesting that pregnancy without complications is possible in patients with NET undergoing PRRT, according to guideline indications, pregnancy is considered an absolute contraindication to PRRT and breast feeding a relative contraindication [28,29].

To date, no human or animal studies on the use of 177Lu-DOTATATE in pregnancy and its effects during breast feeding are available. Nonetheless, for the joint International Atomic Energy Agency (IAEA), European Association of Nuclear Medicine (EANM) and Society of Nuclear Medicine and Molecular Imaging (SNMMI) Guidelines and for the European Neuroendocrine Tumor Society (ENETS) Consensus Guidelines for the Standards of Care in Neuroendocrine Tumors, pregnancy status should be assessed before starting PRRT therapy in women of childbearing age [28,29].

In this regard, effective contraception is recommended both during and for 7 months following the last dose of radionuclide therapy; breastfeeding is also not recommended for 2.5 months after therapy [28,30,31]. For male patients undergoing PRRT with female partners of the reproductive age, the use of effective contraception for 4 months following the treatment is strongly recommended [31]. Guidelines suggest considering sperm banking before therapy due to the temporary impairment of fertility, related to a transient damage to Sertoli cells [28]. On the contrary, no indications are available regarding cryopreservation of oocytes in women [28]. Further investigations are needed to understand the impact of SSA and PRRT on human reproductive function.

#### 3.1.3. Chemotherapy

In recent decades, an increasing amount of evidence has suggested to pay more attention to fertility preservation in cancer patients [32]. National and international guidelines have recommended discussions with patients and their families regarding these issues before the initiation of antineoplastic treatments, incase patients wish to consider a fertility preservation (FP) option [33,34,35]. Indeed, several studies have demonstrated that the conventional treatments for cancer, chemotherapy and radiotherapy can trigger gonadal dysfunction. Thus, gonadal dysfunction can cause premature menopause in women and infertility in both men and women [36,37,38,39,40].

To date, no specific data are available regarding the impact of chemotherapy on fertility in patients with NEN. However, robust data have been published about the independent potential risk of infertility caused by carboplatin and/or cisplatin and etoposide in both males and females [41,42]. These drugs represent the standard of care for patients with NEC, independent of primary tumor origin [3]. Preclinical models have investigated the mechanisms of platinum-based chemotherapy-induced gonadotoxicity [43], demonstrating that chemotherapy damages the reproductive system by enhancing apoptosis [44]. No significant differences in fertility impairments have been detected for cisplatin versus carboplatin [45]. Interestingly, a role for platinum-based chemotherapy-related epigenetic alterations in potentially threatening normal progeny development has been identified [46].

Preclinical studies have highlighted the adverse effects of etoposide on the developing ovaries of female fetuses [47]. Additionally, alkylating agents, such as temozolomide (which is an approved and effective treatment for patients with NET), can impair sperm production in men or deplete the pool of ovarian oocytes in women [48]. To date, the specific effect of temozolomide on reproductive function is still under debate [49]. Changes in semen parameters after the treatment with temozolomide have been observed [50]. However, case reports of both men and women who have produced healthy children after treatment with temozolomide have been published [51].

#### 3.1.4. Targeted Agents

In the era of personalized medicine, targeted agents are largely used across different types of tumors. Two targeted agents have been approved for NETs, the mTOR inhibitor, Everolimus, for lung and gastroenteropancreatic (GEP) NETs or NETs of unknown origin [52,53], and the tyrosine kinase inhibitor (TKI), Sunitinib, for those of pancreatic origin [54]. The toxicity profile of these drugs has been clearly elucidated and detailed in literature while, unfortunately, only scarce data are available on gonadal function and no indication for FP is available (Table 2) [52,53,54].

##### Everolimus

Preclinical data have demonstrated a risk for fertility due to seminiferous tubule dystrophy and reduced tubule diameter after the treatment with Everolimus [55]. Few case reports have shown sperm abnormalities after treatment with the other mTOR inhibitor, Sirolimus, in transplanted patients [56,57]. Other studies revealed significantly lower testosterone levels and a significant increase in gonadotrophic hormones (FSH and LH) in patients treated with Sirolimus [58,59]. Other preclinical studies have suggested a protective role for mTOR inhibitors, leading to the preservation of the ovarian reserve during chemotherapy administration in mice [60,61]. Unfortunately, so far, no clear evidence is available about the impact of Everolimus on cancer patients, including NET patient reproductive function. However, according to the Everolimus Summary of Product Characteristics, women of childbearing potential are advised to use a highly effective method of contraception for the duration of treatment with Everolimus and for up to 8 weeks after the cessation of treatment [62]. This indication suggests that Everolimus could potentially impair cancer patients’ fertility.

##### Sunitinib

Preclinical studies have detected a negative impact of Sunitinib on ovarian function [63]. Specifically, Bernard and colleagues observed a significant impairment in corpora lutea formation in mice receiving Sunitinib compared to control mice, as the consequence of an ovulation defect and/or a luteinization process inhibition. Furthermore, the authors hypothesized that Sunitinib could impact follicular activation and therefore increase atresia [62]. Additionally, Sunitinib has shown to reduce the size of the endometrium in mice [64]. Other studies have demonstrated that Sunitinib had no effects on female and male rat reproduction [65]. Unfortunately, clinical data, in all types of cancer patients, including NET patients, regarding the effects of Sunitinib on gonadal function and subsequent fertility are lacking [66].

### 3.2. NEN Treatment and Sexual Function

Sexual health is defined as “a state of physical, mental, and social well-being in relation to sexuality” [67,68]; however, sexual disorders are often undiscussed during oncology visits [69]. In this regard, cancer and cancer-related treatments could have a detrimental effect on the sexuality of patients and their partners [70]. Stanton et al. highlighted that cancer could affect sexual function, modifying erection, ejaculation and orgasm in male patients as well as arousal, orgasm and satisfaction in females [70]. Therefore, although sexual health should be considered an essential part of QoL in all cancer patients, there are very few studies exploring sexual health in patients with NENs [71,72,73,74,75,76].

Van der Horst- Schrivers et al. explored sexual function in 43 patients with metastatic midgut carcinoid tumors, 27 men and 16 women. Interestingly, the authors observed that male patients with sexual dysfunction showed more long-standing disease as well as a lower tryptophan level. The diagnosis was carried out through the Questionnaire for Screening Sexual Dysfunction (QSD). However, they found sexual dysfunction in 29.6% of men and 6.3% of women, and the prevalence was not higher than in the general population. Considering medical therapy, SSA treatment was not related to a significant difference in QSD score [71].

Similar results were obtained by Zaid et al., who evaluated sexual dysfunctions using the Patient-Reported Outcomes Measurement Information System (PROMIS) on sexual functioning in 57 women, enrolled through social media from eight countries [72]. They found a similar prevalence of dysfunction in the study group compared to the controls. The authors did not observe differences in sexual dysfunction prevalence according to oncological treatment received (surgery alone, radiation alone, chemotherapy alone or associations) [72].

Furthermore, Karppinen et al. evaluated sexual function in 134 patients (74 female and 60 male) with small intestine NET, with a mean disease duration of 81 months; most patients had a metastatic disease, 79% of patients were in treatment with SSA, 26.9% with PRRT and 9.7% receiving other treatments [73]. The diagnosis of sexual dysfunction was carried out through the 15D questionnaire, with only one specific question on sexual activity. Considering this outcome, the authors found worse results in patients compared to the controls [73].

Finally, Talvande et al. described a rare case of a woman with ovarian carcinoid and flushes during coitus [74], while Defeudis et al. reported a rare case of erectile dysfunction after surgery in a patient with Pheochromocytoma [75].

## 4. Conclusions

To date, few data are available on the possible impact of SSA on fertility, mainly based on acromegalic patients. However, pregnancy planning is strongly recommended for women with NETs being treated with SSA, due to the blood–placental barrier crossing of these drugs. Considering the two currently approved targeted agents (Everolimus and Sunitinib), further studies are needed to assess the effects on NET patient fertility, in order to optimize the FP strategy. Furthermore, platinum-based chemotherapy has a well-known negative impact on NEC patient fertility and discussions with patients regarding FP options should be encouraged to improve patient holistic management and QoL. Considering Temozolomide, its effect on reproductive function is still undefined and should be investigated properly, even if changes in semen parameters after treatment have been described. Finally, very few data are available on the sexual function of NEN patients, mainly focusing on the effect of the antitumor treatment. In this regard, further studies are certainly needed, since sexual health should be considered an essential part of QoL.

## Figures and Tables

**Figure 1 jcm-11-03983-f001:**
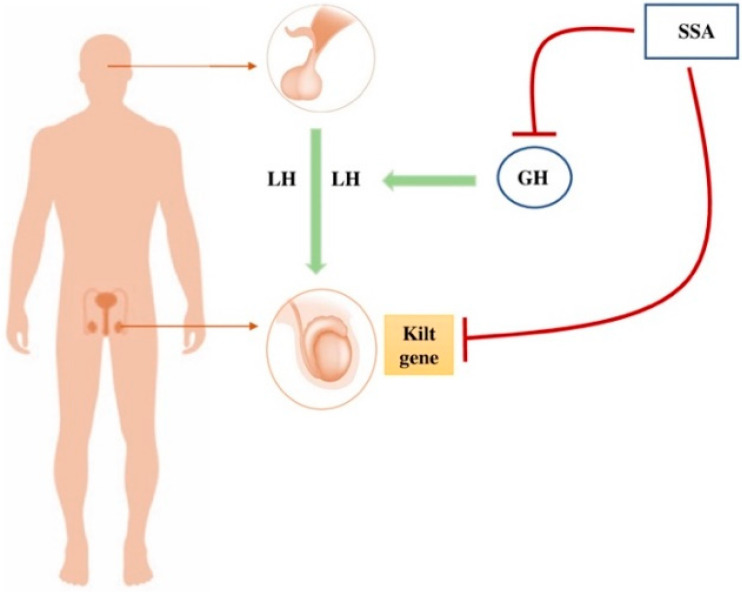
Hypothesis of SSA impact on male fertility. SSA could negatively impact spermatogenesis through two mechanisms. First, an inhibition of GH, and second, a dose-dependent RNA suppression of the kitl gene, involved in the spermatogenesis process. SSA = Somatostatin analogues; LH = Luteinizing hormone; FSH = Follicle-stimulating hormone; GH = Growth hormone.

**Figure 2 jcm-11-03983-f002:**
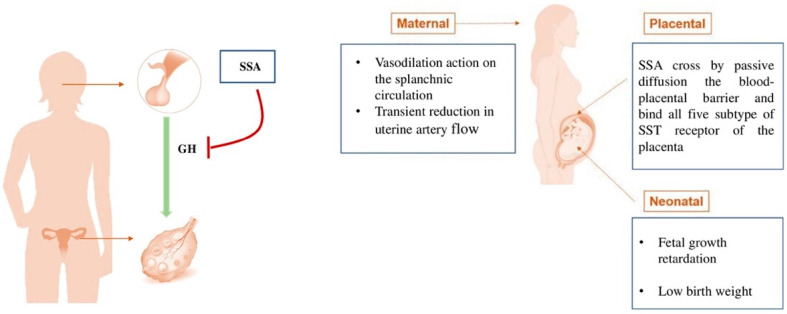
Hypothesis of SSA role in female fertility and pregnancy. SSA could cause a reduction in GH secretion with a negative impact on the growth and onset of fertility. SSA could induce maternal, placental and potentially relevant neonatal effects (as fetal growth retardation and low birth weight). SSA = Somatostatin analogues; GH = Growth hormone.

**Table 1 jcm-11-03983-t001:** Different SSTR subtypes of testicular expressions in humans and animals.

		Cell Types	SSTR1	SSTR2	SSTR3	SSTR4	SSTR5	Methods
**Animal**	**Rat**	Sertoli cells	+	+	+	+	+	q-PCR IHC Western Blot
	**Porcine**	Sertoli cells	+	+	+	-	-	RT-PCR Western Blot
**Human**		Sertoli cells Leydig cells	+	+	-	-	+	IHC

SSTR= Somatostatin receptors; IHC = Immunohistochemistry; qPCR = Quantitative polymerase chain reaction; RT-PCR = Reverse transcription polymerase chain reaction.

**Table 2 jcm-11-03983-t002:** The effect of neuroendocrine neoplasm treatment on male and female fertility.

	Male	Female
**SSA**	A possible detrimental effect due to the induction of Sertoli cell apoptosis.	Blood–placental barrier crossing with possible consequences for offspring (fetal growth retardation and low birth weight).
**PRRT**	A possible transient impairment of spermatogenesis and testosterone reduction.	No sufficient data available.
**Chemotherapy**	Gonadotoxicity by enhancing apoptosis. Impairment regarding sperm production.	Gonadotoxicity by enhancing apoptosis. Depletion of the pool of ovarian oocytes.
**Everolimus**	A possible seminiferous tubule dystrophy and reduced tubule diameter.	No sufficient data available.
**Sunitinib**	No sufficient data available.	A possible ovulation defect and/or a luteinization process inhibition.

SSA = Somatostatin analogues; PRRT = Peptide receptor radionuclide therapy.

## Data Availability

Not applicable.

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
