# Peer review of "Effect of Neuroendocrine Neoplasm Treatment on Human Reproductive Health and Sexual Function"

_jcm, 2022, doi:10.3390/jcm11143983_

Round 1

Reviewer 1 Report

This review describes the effect of NEN treatment on reproductive system and sexual function. This is an important topic and only few literature exists so far.

The review is well written and nicely structured. 

 The authors could additionally include information about recommendations (if any available) regarding:

-      -  fertility preservation options: should this be discussed prior to PRRT?

-       - contraceptive measures during/after PRRT or targeted agents/chemotherapy: How long before attempts to conceive can be made?

Author Response

Thank you for your interesting suggestions. We added the following sentences regarding fertility preservation before chemotherapy: “National and international guidelines have recommended the implementation of the discussion of these issues with the patients and their families before the initiation of antineoplastic treatments to consider a fertility preservation (FP) option. Indeed, several studies have demonstrated that the conventional treatments for cancer, chemotherapy and radiotherapy, can determine gonadal dysfunction. Thus, gonadal dysfunction can cause premature menopause in women and infertility in both men and women”.

No data are available regarding fertility preservation before target agents therapy. We discussed it in the target agents therapy section, as follows: “The toxicity profile of these drugs has been clearly elucidated and detailed in literaturewhile, unfortunately, only scarce data are available on gonadal function and no indication for FP are available”.

About PRRT, we added these sentences: “Despite the Authors of this case report suggested that the pregnancy without complications is possible in patients with NET undergoing PRRT, according to guideline indications, pregnancy is considered as an absolute contraindication to PRRT and breast feeding a relative contraindication.

To date, no human or animal studies on the use of 177Lu- DOTATATE in pregnancy and its effects during breast feeding are available. Anyway, for the joint International Atomic Energy Agency (IAEA), European Association of Nuclear Medicine (EANM) and Society of Nuclear Medicine and Molecular Imaging (SNMMI) Guidelines and for the European Neuroendocrine Tumor Society (ENETS) Consensus Guidelines for the Standards of Care in Neuroendocrine, the pregnancy status should be checked before the PRRT therapy in women of childbearing potential.

In this regard, effective contraception is recommended both during and for 7 months following the last dose of radionuclide therapy; breastfeeding is also not recommended for 2.5 months after therapy. For male undergoing PRRT, in case of female partners of reproductive potential, the use of effective contraception for 4 months following the treatment is strongly recommended.

Guidelines suggest considering sperm banking before therapy due to the temporary impairment of fertility, related to a transient damage to Sertoli cells. On the contrary, guidelines no indications are available regarding cryopreservation of oocytes in women.”

Reviewer 2 Report

This manuscript reviews the negative effect of treatment to neuroendocrine tumor on sexual function. Limited data have been reported in this field. The authors well documented the effect and mechanism of NET treatment on reproductive organs. I would recommend the authors to add a figure which visually shows the mechanism or hypothesis of negative reproductive effect of somatostatin analogues.

Author Response

R: Thank you for your suggestion. We added figures which visually shows the mechanism of negative reproductive effect of somatostatin analogues both in male and female.